# Old Tricks, New Opportunities: How Companies Violate the International Code of Marketing of Breast-Milk Substitutes and Undermine Maternal and Child Health during the COVID-19 Pandemic

**DOI:** 10.3390/ijerph18052381

**Published:** 2021-03-01

**Authors:** Constance Ching, Paul Zambrano, Tuan T. Nguyen, Manisha Tharaney, Maurice Gerald Zafimanjaka, Roger Mathisen

**Affiliations:** 1Alive & Thrive Southeast Asia/FHI 360, Washington, DC 20009, USA; 2Alive & Thrive Southeast Asia/FHI 360, Manila 1101, Philippines; pzambrano@fhi360.org; 3Alive & Thrive Southeast Asia/FHI 360, Hanoi 10000, Vietnam; tnguyen@fhi360.org (T.T.N.); rmathisen@fhi360.org (R.M.); 4Alive & Thrive West Africa/FHI 360, Abidjan, Côte d'lvoire; mtharaney@fhi360.org; 5Alive & Thrive West Africa/FHI 360, Ouagadougou, Burkina Faso; mzafimanjaka@fhi360.org

**Keywords:** breastmilk substitutes (BMS), International Code, breastfeeding, aggressive marketing, baby formula, COVID-19, emergencies, maternal child health, infant and young child feeding, malnutrition

## Abstract

Breastfeeding is critical to maternal and child health and survival, and the benefits persist until later in life. Inappropriate marketing of breastmilk substitutes (BMS), feeding bottles, and teats threatens the enabling environment of breastfeeding, and exacerbates child mortality, morbidity, and malnutrition, especially in the context of COVID-19. These tactics also violate the International Code of Marketing of Breast-Milk Substitutes. This study identified marketing tactics of BMS companies since the start of the COVID-19 pandemic by reviewing promotional materials and activities from 9 companies in 14 countries, and the official Code reporting data from the Philippines. Eight qualitative themes emerged that indicate companies are capitalizing on fear related to COVID-19 by using health claims and misinformation about breastfeeding. Other promotional tactics such as donations and services were used to harness the public sentiment of hope and solidarity. Past studies show that these tactics are not new, but the pandemic has provided a new entry point, helped along by the unprecedented boom in digital marketing. There was a sharp increase of reported violations in the Philippines since the pandemic: 291 during the first months of the outbreak compared with 70 in all of 2019, corroborating the thematic findings. A lack of public awareness about the harm of donations and inadequate Code implementation and enforcement have exacerbated these problems. Proposed immediate action includes using monitoring findings to inform World Health Assembly (WHA) actions, targeted enforcement, and addressing misinformation about breastfeeding in the context of COVID-19. Longer-term action includes holding social media platforms accountable, raising public awareness on the Code, and mobilizing community monitoring.

## 1. Introduction

Breastfeeding is the biological norm [1]. With its nutritional, immunological, neurological, endocrinological, and ecological benefits, it is critical to maternal and child health and survival [2]. Breastfeeding protects against short-term morbidity and mortality due to infectious diseases and long-term chronic diseases including obesity and diabetes [3]. It also has human capital and socio-economic benefits [4]. However, it is a complex health behavior, and public health policies, maternity protection, workplace culture, and market forces are some of the structural factors that influence breastfeeding practices [2,5]. Aggressive marketing of breastmilk substitutes, bottles, and teats (hereinafter collectively referred to as “BMS”) contributes to suboptimal breastfeeding and remains as a major structural factor that threatens the enabling environment of breastfeeding [2]. It misleads the public to accept bottle-feeding as the social norm, changing infant feeding from a dynamic maternal and child relationship to a commodity [1,6,7].

### Breastfeeding and Its Challenges during the COVID-19 Pandemic

Breastfeeding and its nutritional and protective health benefits are particularly important in the context of COVID-19 [8,9], as the pandemic is straining health care systems and increasing food insecurity, especially in low- and middle-income countries [10]. A recent study found breastfeeding to have a significant relationship with lower COVID-19 infection risk [11]. Even though there is currently no evidence that COVID-19 is transmitted via breastfeeding [12,13] and the World Health Organization’s (WHO) guidance supports breastfeeding with appropriate precautions for mothers with suspected or confirmed COVID-19 [14], the pandemic is threatening breastfeeding practices. Depleted healthcare resources and reduced use of health facilities are compromising antenatal and postnatal care and lactation support [15]. COVID-19 guidelines in several countries discourage breastfeeding [12,16,17] and some studies have raised questions about breastfeeding in the context of COVID-19 [18,19,20]. BMS manufacturers and distributors (including bottles and teats companies) are aggressively marketing their products in violation of the International Code of Marketing of Breast-Milk Substitutes [21,22,23,24]. These marketing practices exacerbate disruptions in breastfeeding, and are likely to have compounding effects at a time when risks of maternal and child mortality, morbidity, and malnutrition are already heightened [10,15,25].

## 2. Marketing and Breastfeeding

### 2.1. The International Code of Marketing of Breast-Milk Substitutes

In the 1970s, there was a global recognition that unregulated marketing and inappropriate use of infant formula contributed to an alarming decline in breastfeeding and widespread infant malnutrition and mortality [26]. In response to this decline, the International Code of Marketing of Breast-Milk Substitutes was adopted by the World Health Assembly (WHA) in 1981 with the aim to protect breastfeeding [27]. Its ultimate purpose is to restrict all forms of commercial promotion that undermine breastfeeding by outlining the responsibilities of governments, healthcare systems and workers, and BMS companies. It also minimizes health risks for formula-fed babies by requiring appropriate information on health hazards and proper use and preparation. Subsequent relevant WHA resolutions keep the International Code of Marketing of Breast-Milk Substitutes updated with evolving marketing trends and the latest global health recommendations (hereinafter the International Code of Marketing of Breast-Milk Substitutes and relevant WHA resolutions are jointly referred to as “the Code”) [27,28]. Governments have the moral and political obligation to adopt the Code into national legislation. The Code also has human rights implications, as breastfeeding is central to the right of the child to the highest attainable standard of health (Article 24) enshrined in the Convention on the Rights of the Child (CRC) [29]. In 2013, the CRC committee released General Comments that highlight the obligation of State Parties to adopt legislation based on the Code and for industry to fully comply with it [30] (see Appendix B & Appendix C for overview of the Code and summary of all relevant WHA resolutions up until 2018).

### 2.2. How Do BMS Companies Promote?

Forty years after the adoption of the Code, the problems of both rampant marketing and malnutrition persist through the “industrial epidemic” of infant and young child feeding [6]. Inappropriate BMS marketing tactics come in different forms, including direct promotion to consumers (TV advertisements, home visits, or sending samples through the mail), to (and through) health professionals and systems (offering stipends to attend sponsored meetings, free gifts with company logos, and providing free formula in hospital maternity discharge packs), and to policymakers (lobbying with government’s trade and commerce departments by industry) [5]. A review of Code violations in the triennial series of *Breaking the Rules, Stretching the Rules* by IBFAN-ICDC since 1982 showed that Code violations continue to be widespread globally. In the 2014–2017 report, there were more than 800 violations documented on 28 baby food companies, including bottles and teats companies, from 79 countries [31].

### 2.3. Aggressive Marketing during COVID-19

BMS companies’ promotional activities are exploiting the COVID-19 pandemic. Several examples are documented on the Baby Milk Action’s website [23], including Danone using social media to contact mothers in India, and Nestlé donating Lactogrow, a growing-up milk (hereinafter known as GUM), to the Provincial Disaster Management Authority in Pakistan. BMS companies are donating to hospitals, governmental agencies, and non-governmental organizations (NGOs) on a global scale [22,24]. A recent online survey in the United Kingdom reported that 80 percent of the 1360 breastfeeding mothers interviewed had been contacted by formula companies, typically on social media [24]. These reports provide context for the current study that further examined how marketing tactics have been employed by companies to promote their products and undermine breastfeeding.

## 3. Aim, Materials and Methods

The aim of this study is to examine the marketing tactics of BMS companies since the start of the COVID-19 pandemic. This study seeks to address the following research questions:Are BMS companies marketing aggressively during the COVID-19 pandemic?If so, how are BMS companies capitalizing on the pandemic for their marketing activities?What are the implications for policy and programs?

### 3.1. Data Collection and Analysis

The research uses a concurrent mixed nested design (QUALquan) with a predominant qualitative method and an embedded quantitative strand.

#### 3.1.1. Qualitative Data 

Promotional activities and materials by BMS companies dating from 30 January 2020, when the WHO declared COVID-19 a “public health emergency of international concern”, were identified as appropriate data for this study. Purposive sampling was used to collect data via the internet (e.g., infant feeding and child nutrition blogs, social media, company websites), print magazines, networks of health officials and professionals, health facilities, and shops from August to October 2020. The research team oversaw data collection, and country focal points (health workers and professionals) were recruited through the authors’ existing professional networks to support data collection in their respective countries. Among all the collected data, only examples that made direct or indirect references to or coincided with the COVID-19 pandemic were selected. Twenty-six examples from 9 companies in 14 countries (Burkina Faso, Canada, China, India, Indonesia, Kenya, Laos, Malaysia, Myanmar, Pakistan, Singapore, The Philippines, United States, and Vietnam) found via the internet and in a print magazine were selected for further analysis. Country focal points and the corresponding author assisted in translating materials in Chinese, Burmese, French, Indonesian, Lao, and Vietnamese to English. Full descriptions of each selected piece of material were then transcribed into English.

The study used thematic analysis to identify and describe categories and themes [32,33]. The first author was the primary coder in charge of transcribing the materials and overseeing the coding process. To strengthen inter-coder reliability, two co-authors reviewed the selected data and the coding structure completed by the primary coder. Discrepancies were reviewed and consensus was negotiated among the research team to establish a single application of final codes [34,35]. A debriefing (review of data and research process) with a professional peer who is familiar with the topic of research was conducted to establish validity [36].

A constant comparison technique was used to inductively compare emerging categories with previous ones, and data were gathered continuously and simultaneously with analysis until saturation [37]. First-level coding was conducted to separate the data into different categories based on their manifest content [38], such as the type of promotion, wording, and catch phrases used, and the nature of the activity (e.g., claims on immunity, donations, and discounts). Second-level coding involved extrapolating latent content from the categories to identify developing themes such as using immunity claims to respond to fear of transmission, or donations to health workers to show solidarity. To identify the underlying message and aim, questions such as “What is the main message(s) this social media post is trying to communicate with the audience? What is this advertisement using to make parents want to buy the product? How is this promotional? How are the perceived messages connected to the COVID-19 pandemic?” were used to delineate more abstract messages and concepts. Different themes were then compared and grouped into broader descriptive themes.

#### 3.1.2. Desk Review with Data from the Department of Health Philippines

An official database of reported violations of Executive Order 51 (the Milk Code) in the Philippines (data from January 2019 to July 2020) from the Department of Health Philippines was obtained in August 2020. It was the only systematically-collected official dataset on national Code violations the research team could obtain. The reports in the database were obtained through a crowd-sourced reporting system that utilizes mobile and web-based platforms with standardized form fields. The data included information about violators, product types, type of violations, and channels. Using the database, descriptive information about violations reported by date and type was extracted.

## 4. Results

### 4.1. Qualitative Themes

Eight broad themes emerged from the selected data after thematic analysis. Table 1 below provides a brief summary.

#### 4.1.1. (i) Unfounded Health Claims on Immunity That Prompt Fear

BMS manufacturers and distributors have directed their marketing activities to emphasize immunity and resilience, sometimes explicitly or implicitly referring to COVID-19. This gives the impression that BMS products can help combat COVID-19 in babies or make them less susceptible to transmission.

In Vietnam, Colosbaby (Appendix A & Figure 1) [39,40], BMS manufactured by Vitadairy, claimed the product can “boost immune system, prevent respiratory, and digestive infections caused by viruses and bacteria”, and help babies to “gain weight healthily, avoid constipation, sleep well and grow rapidly, and increase height”. The brand appropriates colostrum by using “Colos” in its name, and claimed to provide “Colos IgG” (immunoglobulin G), the most prominent antibody found in colostrum, humanizing the product and making it comparable to breastmilk.

In Pakistan, Nestlé used the high number of confirmed COVID-19 cases as a selling point. A series of short videos have appeared on Nangrow Pakistan’s Facebook page showing a toddler, each with taglines that make an indirect reference to COVID-19, asking parents whether they are worried about their child, how they are preparing their child for the world outside, and how Nangrow’s HMO (human milk oligosaccharides) can help boost their immunity (Figure 2) [41,42,43].

#### 4.1.2. (ii) Riding on Public Health Authority to Gain Legitimacy

In the Colosbaby Facebook post in Vietnam, an advertisement featured a headshot of WHO Director-General Dr Tedros Adhanom Ghebreyesus, with the captions: “WHO raises COVID-19 threat warning to its highest level” and “The world is entering a decisive moment when the coronavirus disease is spreading rapidly” (Figure 2) [40]. The statements explicitly linked the product’s immunological claims to the fight against COVID-19, which created an impression that the product can help to fight the virus. The unauthorized use of a picture of the WHO Director-General falsely represented the product as being endorsed by the highest authority on health, which was an attempt to align the product with an international public health establishment to gain legitimacy.

#### 4.1.3. (iii) Appealing to Public Sentiment on Solidarity and Hope

As opposed to capitalizing on fear, companies like Mead Johnson (owned by Reckitt Benckiser), Abbott, and FrieslandCampina in China captured the public sentiment of hope and resilience. Being the first country to endure the outbreak, these companies used cause marketing campaigns to convey goodwill, resonating with the hope of containing the COVID-19 outbreak, and for life to return to normal. Through campaigns with slogans such as “Conquer virus for love to protect new lives” (Mead Johnson) (Figure 3) [44], “A warm heart to keep you company and to carry on with the love of a mother: Abbott fully supports mothers returning to work” (Abbott) (Figure 4) [45], and “Wuhan add oil (keep it up), united with a single will strong like a fortress, we can get through the challenges together” (FrieslandCampina) (Appendix A) [46], they aligned themselves as a united front with the public, appealing to their sense of solidarity during the global health crisis. These campaigns were fraught with donations of “nutritional packages” that contain BMS, milk for mothers, and offering of services that sought to contact mothers or pregnant women.

Being in partnerships with charitable foundations including China Children and Teenagers’ Fund (Mead Johnson), Chinese Red Cross Foundation (Abbott), and China Women’s Development Foundation (FrieslandCampina), these campaigns optimized the ‘power of association’ to legitimize themselves as non-profit driven. By linking their products to charitable causes through dissemination of samples, donations, and offering of services, these campaigns appeared as charitable acts rather than marketing tactics to induce sales.

#### 4.1.4. (iv) Influx of Donations of BMS and Supplies Related to COVID-19

In China, Abbott provided ‘donation packets’ containing Similac and Eleva premium GUM products to the Chinese Red Cross Foundation to be redistributed to mothers returning to work (Figure 4) [45]. The caption “Abbott fully supports mothers returning to work” appeared on the e-poster, in which two mask-wearing mothers were holding hands with two hearts next to each other. Although presented as donations, these were essentially premium sample packs of BMS for promotional purposes targeting professional mothers with purchasing power.

Using the excuse to prevent stunting, Danone in Indonesia donated SGM and Bebelac GUMs and milk for mothers to the Central Java government, targeting pregnant and lactating mothers (Figure 5) [47]. Meanwhile, in Canada, Nestlé has provided food donations, including “baby/infant food” specifically for “families and communities impacted by COVID-19”, to Food Banks Canada, a national organization that works in partnership with other organizations at a community level to address hunger [48]. In the Philippines, out of the 291 Code violations reported to the Department of Health since the COVID-19 outbreak, a significantly high proportion of them, 81 percent (235 cases), were related to BMS donations [49].

Apart from BMS, companies also donated food items, hygiene kits, personal protective equipment (PPE), and medical equipment to healthcare facilities and health workers. In partnership with the Chinese Red Cross Foundation, Feihe, a Chinese BMS manufacturer, donated RMB30 million to help build a hospital in Wuhan, negative-pressure ambulances for COVID patients, PPE, and remote fetal heart-rate monitors to hospitals in Hubei for pregnant women (Appendix A) [50]. In Malaysia, Danone donated masks and hand sanitizers to 4000 nurses at 50 government health clinics across the country [51]. In Burkina Faso, Nestlé donated food and beverages to Ministère de la Femme, Solidarité nationale, Famille et de l’Action Humanitaire (Ministry of Women, National Solidarity and Family, and Ministry of Humanitarian Action) to be distributed to internally-displaced persons, breastfeeding women, and children (Figure 6) [52]. Additionally, respirators, hygiene kits, and surgical masks were donated to the Ministry of Health during the COVID-19 pandemic (Appendix A) [53].

The partnerships with charitable foundations or governments helped launder the donations through a third party, legitimizing company promotional tactics as charitable or public campaigns rather than marketing schemes. 

#### 4.1.5. (v) Prominent Use of Digital Platforms to Reach Out to Parents

Social distancing and varying degrees of movement restrictions have led to the increased demand for medical consultation. The pandemic has given practices such as telemedicine a rising platform to popularity. In China, Mead Johnson, manufacturer of Enfamil A+, Enfagrow A+, and premium brand Enfinitas, partnered with the country’s largest telemedicine platform, Chunyu Yisheng (www.chunyuyisheng.com, accessed on 1 September 2020), to offer psychological recovery counseling services and online medical consultations specifically for pregnant women and new mothers (Figure 3) [44]. It also hired infant and young child nutritionists to provide online workshops for mothers on information related to COVID-19. Feihe, a top-selling Chinese formula manufacturer, also offered free online medical consultation as the company spokesperson described the pandemic as a marketing opportunity for the brand [54].

In Indonesia, Nutricia (owned by Danone), manufacturer of Nutrilon and Bebelac, offered a 24/7 hotline service during Eid Al-Fitr (Muslim religious holiday) through its careline Nutriclub (Figure 7) [55]. The tagline “we’re standing by you, let’s stay resilient together” was used in an Instagram post as a reference to the physical distance policy that imposed traveling restrictions during the holiday. While the tagline was an indirect reminder of the social isolation from friends and family, it also suggested solidarity with parents amid the pandemic. As part of the purported message, the post told parents “not to worry” as it iterated Danone’s commitment to providing parents with important child nutrition information.

Dugro (owned by Danone) in Myanmar partnered with a popular telemedicine platform OnDoctor App and an influencer nutritionist Dr Jasmine on a Facebook vlog series that target parents (Appendix A) [56]. The series addressed infant and young child nutrition, with the Dugro 3 logo appearing on the corner of the screen, with the hashtags #DugroMyanmar, #DumexDugro, and #WellroundedKid. The videos are accessible on the social media accounts of OnDoctor App (with over 1.5 million followers) and Dr Jasmine. Dugro also targeted another popular social media platform, the Myanmar Parenting Group on Facebook (Appendix A) [57]. With over 40,000 active members, the online peer support group adorned a Dugro 3 banner with a child holding a glass of milk and a tagline “get this to make it more complete, and become excellent and smart babies”. The Dugro banner only appeared in the group in July 2020, a few months into the COVID-19 pandemic, and it came with a hefty sum of sponsorship money (T. Thien, personal communication, 8 September 2020).

#### 4.1.6. (vi) Promoting Uncertainty Through Endorsing Breastfeeding

In Singapore, Illuma (owned by Nestlé) invested in a full-page advertisement in a mother’s magazine dedicated to the benefits of breastfeeding (Figure 8) [58]. The content appeared to be an informative article, with advice from a breastfeeding specialist on breastfeeding topics such as “How can I protect my newborn from COVID-19?” and “How long should I breastfeed my child for it to be beneficial?” The information made references to breastfeeding and the properties “sn-2” and “A2 beta-casein”, and statements such as “nutrient absorption”, “strong bone”, and “support gastrointestinal well-being”. Yet, paradoxically, next to that was a full-page advertisement of Wyeth’s Illuma 3, described as “human affinity”. The page featured benefit icons “sn-2” and “A2 beta-casein”, the same properties referenced in the page about benefits of breastfeeding, and a tagline, “the only formula in Singapore with 5 benefits of Sn-2”. When reading the two pages together, it became clear that Nestlé was allying its product with the superiority of breastfeeding, and then stressing the product’s similarity to breastmilk. Thus, the more it promoted breastfeeding, the more their product was promoted.

In India, bottle and teat company, Pigeon, published a video on social media through its #standbynursingmoms campaign to promote breastfeeding and to support breastfeeding week during the pandemic (Appendix A and Figure 9) [59,60,61]. The text that introduced the campaign described breastfeeding as a beautiful experience and praised the importance of breastfeeding, but sentences like “nursing mothers who go through an emotional roller coaster ride and uncertainty…” and “nursing moms go through pangs of anxiety and discomfort …” were found later in the text. In a post about the campaign, the tagline started with “we know breastfeeding is not easy…”, while in a video, it asked mothers to reveal their “honest feelings” about breastfeeding by showing a cue card. Among the positive descriptions, there were cards showing words like “uncomfortable” and “exhausting”. 

In the United States, Medela, a company that manufactures breast pumps and bottles and teats, provided a free webinar on “Supporting Breastfeeding During COVID-19” (Appendix A) [62]. The webinar’s description posed anxiety-provoking questions such as “Will I be able to be with my baby if I test positive? Will I be able to breastfeed and get the support I need?” It mentioned “evolving recommendations about postpartum care for the breastfeeding mother/baby dyad” without directly stating that breastfeeding is safe and recommended in the context of COVID-19. It addressed the importance of “breastmilk” during difficult times, stating “all major organizations agree that the provision of breastmilk is important and should be a priority in this pandemic”, but omitted breastfeeding itself. Medela in Kenya also posted on its Facebook page “Coronavirus: Information for breastfeeding moms and moms to be”, linking the virus with breastfeeding, and posing the question “Is it safe to breastfeed if I have Coronavirus?” (Figure 10) [63]. The answer provided in the caption avoided addressing breastfeeding directly, and instead emphasized the mother continuing “to give her milk to the baby”. The omission of breastfeeding serves the interest of a company that benefits from bottle-feeding. The caption included a hashtag “#MedelaBreastPumps”, conveniently leading parents to promotion about Medela infant feeding products.

At a time of contradictory health messages and practices that confuse the public, the Hi-Family Club Facebook page (sponsored by Nutricia, Danone) in Laos posted a video that appeared to provide information on breastfeeding (Appendix A) [64]. The video linked breastfeeding with questions such as “Do you wonder why your breastmilk color is different sometimes? Check what each color of milk indicates and whether it is harmful to the baby. Watch the video and see more details about breastfeeding concerns here”. Linking different breastmilk colors and questioning whether it is harmful to one’s baby can provoke fear in mothers, shaking their confidence in breastfeeding. A breast-pump appeared in the video that further removed the idea of breastfeeding from the ‘feeding norms’, and the benefit icon at the bottom was referring to proprietary products.

#### 4.1.7. (vii) Discounts on BMS Products That Are Linked to COVID-19

Many families have experienced financial and emotional hardships during the COVID-19 pandemic. During health crises, consumers sometimes panic about a shortage of retail supply. Companies capitalize on this by providing COVID-19-related health information and messages to reassure customers that there will be enough supply. Discounts, a very direct form of promotion, are also offered and are disguised as a caring act toward their customers during a difficult time. In the United States, the Gerber website (owned by Nestlé) had a page that was dedicated to “an important message for the Gerber Community on COVID-19”, reassuring customers there is adequate supply and clarifying false rumors on free giveaways (Appendix A) [65]. However, on the same page, discount coupons were offered, stating “we are committed to our goal of supporting babies and families with dependable, affordable nutrition …”. Similac (owned by Abbott), on its own company website, provided health information on the COVID-19 pandemic and reminded its customers that “in difficult times, count on Similac … You promise to support and nourish your baby, we promise to help”. Discounts and gifts were offered as “a little support can make all the difference …, difficult times can create a financial burden” (Appendix A) [66].

#### 4.1.8. (viii) Reaching Out to Health Professionals through Sponsoring Educational Events on Topics Relating to COVID-19 and Infant and Young Child Feeding

Company sponsorship of educational events for health workers can be seen as an endorsement by the medical establishment. In Kenya, the infant formula industry has accelerated efforts to target professional associations during COVID-19. Nestlé reached out to the Kenya Pediatric Nurses Chapter, the Nutritionists Association of Kenya, and several health workers from prominent hospitals in Nairobi, and provided them with information sessions on topics such as “the burden of micronutrient deficiencies in infancy and the unsuitable use of BMS” (J. Kavle, personal communication, 5 September 2020). Ministry of Health Kenya recognizes that such tactics provide an open channel for companies to discuss with health professionals how BMS can be ‘suitable’, especially in the context of COVID-19, serving the companies’ interest (Appendix A) [67]. In the Philippines, Abbott and the Medical College of the University of the Philippines organized the webinar “Pregnancy in time of COVID-19”, targeting health workers (Figure 11) [68]. These sponsored events are fraught with conflicts of interest, potentially allowing companies the chance to influence the information on topics such as infant nutrition and breastfeeding in the context of COVID-19.

### 4.2. Reported Violations during COVID-19 in the Philippines

Since the onset of the outbreak, there have been rampant donations of BMS at municipal level in the Philippines [69]. This is supported by data contained in the official national database of reported Code violations or aggressive marketing activities from January 2019 to July 2020 [49]. The government declared a state of public health emergency and imposed strict social distancing measures in early March 2020 and, from March to July 2020, there were a total of 291 reported violations, a sharp increase from previous months. In the whole of 2019, only 70 complaints were received, and one reported case in January 2020 (Figure 12).

Out of the 291 cases reported since the COVID-19 outbreak, a significantly high proportion of them, 81 percent (235 cases), were related to donations of BMS. Twenty-four of the reported cases were related to sponsorship, nine to mass media promotion, six to health workers, five reports on gifts, four reported cases on promotion in shops, one report on promotion in the health system, and seven were classified as others (Figure 13). In 2019, there was only one reported case of donation, indicating a huge increase in donations since the start of the COVID-19 pandemic.

While many of the donations reported were requested or organized by individuals and groups that appeared to be self-initiated, some were solicited or organized by local politicians, celebrities, and radio stations. The findings indicate a potential lack of public awareness on the risks of BMS donations in emergencies, as well as an overall confusion about restrictions on donations at the local government level [69].

## 5. Discussion

### 5.1. Overview of Findings

The central themes that have emerged in this qualitative analysis indicate BMS companies have tailored their marketing activities toward exploiting opportunities related to the COVID-19 pandemic (Table 1). The quantitative findings indicate a sharp increase in inappropriate marketing activities in the Philippines in parallel with the onset of the COVID-19 outbreak (Figure 12), which corroborates the thematic findings; especially apparent is the high number of donations among the reported cases since the start of the outbreak.

### 5.2. Old Tricks That Have Found New Opportunities

The results of this study are not surprising, but are highly alarming. Past examples in this section are used to contextualize the findings, confirming that these tactics used by BMS companies are not new, but the COVID-19 pandemic has provided them with a new platform for promotion.

Health claims have been a prime promotional tool to increase market value for many years, even though they are prohibited by resolution WHA 58.32 (2005) [28]. These products with complicated additives assert a wide range of health, cognitive, and developmental benefits including enhancing intelligence, visual acuity, digestive health, ability to sleep well, and immunity [31]. These claims often make their additives comparable to properties found in breastmilk to mislead parents. The claims to various health benefits are often inadequately supported and, in some cases, unfounded [70,71,72]. Some health claims also appear in trademarked logos or icons as attempts to circumvent the Code [31].

Companies sometimes align their products with public health establishments and authorities to gain trust from the public. They also carefully present the products as only second best by emphasizing “breast is best” to not appear to be competing with breastfeeding [6]. Nestlé China turned the United Nations International Children's Emergency Fund’s (UNICEF) 1000 days campaign into the “excellent care for 1000 days, excellent lifetime protection” campaign, which was craftily combined with Nestlé’s signature “Start Healthy, Stay Healthy” slogan [31].

Donations create a culture of dependency and an obligation of reciprocation, which can adversely affect the promotion of breastfeeding [30]. Resolution WHA 47.5 (1994) prohibits BMS donations in the health systems. Resolution WHA63.23 (2010) calls on governments to adhere to the Operational Guidance on Infant and Young Child Feeding in Emergencies [28,73], which stipulates that donations of BMS, complementary foods and feeding equipment should not be sought or accepted, and supplies should be procured through official channels. However, as a way to widen existing markets and facilitate public relations, companies have long capitalized on public health emergencies caused by natural hazards, internal conflicts, epidemics, and wars to give donations [25,74,75,76,77,78]. The harmful impact of BMS donations in emergency contexts has been well-documented [74,75,76,77,78]. The unsolicited BMS, and the indiscriminate distribution of them, can easily spillover to breastfeeding mothers, potentially resulting in mixed-feeding or cessation of breastfeeding. This in turn increases the rates of artificial feeding and inevitably contributes to child morbidity and mortality [75,78]. Even with infants and children for whom breastfeeding is not possible, such distributions do not necessarily support their survival. It is difficult to ensure proper usage, and other resources necessary for safe formula-feeding (e.g., clean water) are not available in emergency situations [75]. Donations often include other products such as milk for mothers—although not covered in the scope of the Code, these products create a false dependence that undermines women’s confidence in breastfeeding.

Nestlé had capitalized on influencer marketing through their partnership with socialsoup.com, using a peer-to-peer influencing strategy to recruit mothers to test baby food products and post pictures and videos on social media. The tactic led to mothers becoming their primary promoters. The promotion effect cascaded through each of their social media networks and multiplied like a pyramid scheme. Their posts were fed back into live campaign hubs and the Nestlé Australia website. Over 25,000 mothers became their brand ambassadors within a few years [31]. BMS companies have been specializing in soft-selling techniques by building faux, but long-term relationships through online activities like carelines, brand ambassador marketing and baby clubs [6].

Publicly endorsing and promoting breastfeeding has become a popular tactic among BMS companies in recent years. Being proactively breastfeeding-friendly is also an effective way to ingratiate the company’s name among health professionals and mothers. In Chile, Nestlé published its own Breastfeeding Manual, and in Gabon, Nestlé’s pamphlet on breastfeeding was distributed to mothers in health facilities and several professional associations [31]. Companies may initially use breastfeeding-friendly language and appear as genuinely supportive, but in the tips and recommendations lie the subtexts (e.g. over-emphasizing difficulties and how to solve them). Although they do not always explicitly promote BMS products, messages that are of subpar information, in the form of questions or simply unclear can sufficiently instill doubts, influencing mothers’ confidence and hence decisions toward breastfeeding. Often, the kind of ‘pseudo information’ that appears on social media falls through the cracks of the laws, owing to its nebulous nature. In the context of COVID-19, BMS companies can exploit the existing misinformation on breastfeeding to justify BMS donations. This is extremely risky, as the early response to the global HIV epidemic has shown that efforts to prevent vertical transmission by replacing breastfeeding with formula feeding ultimately resulted in more infant deaths [22]. Company messages that covertly question the safety and desirability of breastfeeding, together with the scientific studies and clinical practices that cast doubt on it, can seriously undermine current WHO’s and UNICEF’s recommendations on breastfeeding in the context of COVID-19.

Companies are eager to offer sponsorship for conferences and research, and forge strong financial links with medical establishments, professional associations, and public health agencies [6]. Currently, 38 percent of national pediatric associations receive funding for their conferences from BMS manufacturers [79]. The bulk of marketing activities are targeted at health workers and health systems [79], because they are the primary source of trusted advice and have immense influence over infant and young child feeding decisions. For example, the Kartini Program in Indonesia is a government program sponsored by Nestlé to train midwives to support mothers on exclusive breastfeeding [31]. The Nestlé sponsorship may explain the program’s focus on reaching new mothers who may not have established confidence in their breastfeeding practices, and its lack of emphasis on continued breastfeeding. Resolution WHA 58.32 (2005) prohibits incentives for health professionals that create conflicts of interest [28]. The WHO Guidance on Ending the Inappropriate Promotion of Foods for Infants and Young Children (hereinafter known as the 2016 Guidance) [80] states that companies are not allowed to sponsor meetings or provide incentives to health professionals and workers. However, many health workers are unaware of any conflict of interest, especially when the events are framed in ways that are ‘health-giving’ [81,82]. The fiduciary duty of health workers and the BMS companies’ primary interest are in direct conflict with each other. Conflicts of interest arise whenever there are sponsorships or partnerships that bind these two conflicting duties or interests together. Moreover, any visible links between health workers and the company may send a message of positive endorsement in the eye of the public. Often, companies claim these sponsorships and partnerships as their corporate social responsibility, which is yet another way to promote public image while influencing healthcare practices, programs, and in some cases policy. However, the Global Strategy for Infant and Young Child Feeding, endorsed by resolution WHA55.25 (2002), clearly states that the role of companies is only confined to full compliance with the Code and ensuring their products’ quality, safety, and labeling meet the Codex Alimentarius standards [30].

The BMS industry often makes efforts to influence infant feeding policy [5], even though there are resolutions such as WHA65.6 (2012) that urges safeguards to be established against conflicts of interest in nutrition action [28]. For instance, BMS industry was behind the efforts to try to amend the Milk Code in the Philippines in 2012. Apart from attempts to lift the ban on BMS donations in emergencies and BMS samples to be distributed in healthcare facilities, the BMS industry also attempted to reduce the scope that covered up to 36 months to 6 months of age, and to allow BMS company staff to be involved in breastfeeding activities [83,84]. In public health, policymakers’ and governments’ primary duty is the furtherance of health and nutrition outcomes. Any form of BMS industry’s involvement or partnership that influences policy creates conflicts of interest [85].

### 5.3. Where Is the Accountabilty from Social Media?

The growing concern over the widespread use of digital platforms as part of marketing strategies has been reflected in several WHA Resolutions for the past two decades—as early as 2001 and as recent as 2020 [28,86]. Our findings, most of which were found on the internet, show that promotion of BMS has found even newer heights through the virality and omnipresence of digital marketing. Now, amid the COVID-19 pandemic, dependence on digital platforms for daily living needs has grown exponentially.

Over time, online promotions are becoming increasingly bespoke with interest-targeting tactics and web-tracking tools [6,87]. Social media such as Facebook and Instagram use data mining to collect and analyze consumer information, and algorithms to optimize marketing capacity and acuity. Company-sponsored influencer blogs or vlogs and online social groups can go viral on social media with features such as follows, shares, likes, and hashtags. The ‘mombassador’ phenomenon [21], in which companies like Danone and Nestlé use social media and mothers to promote baby formula, suggests that the social and commercial boundaries have been blurred further. Parents are increasingly vulnerable to the predatory marketing that can alter their decisions on how to feed their children. While BMS companies are rightfully blamed for being the culprits that exploit social media for their predatory marketing, media platforms that host countless inappropriate marketing activities are often being seen as neutral. The weight of the Code as a global recommendation that protects child health means that social media’s role as an accomplice to predatory marketing should not be seen as so innocuous. Instead, they should also be held accountable to comply with the Code.

### 5.4. Gaps and Challenges in Code Implementation

Code implementation remains as the most effective mechanism to curb inappropriate marketing and protect breastfeeding—given that it is legislated in its entirety (incorporating all relevant WHA resolutions), properly enforced, and continuously and sustainably monitored. To date, only 25 countries out of 194 countries have measures that are substantially aligned with the Code, while 58 have no legal provisions [79].

Some of the Code violations in our findings are from countries where there are no Code-based legal measures, such as Canada, Malaysia, Singapore and the United States [79]. As for China, the Code measure adopted in 1995 was repealed in 2016 without any replacement, allowing companies to openly reach out to mothers via various online platforms [79,88].

Some findings are from countries where the laws are unable to capture the wide range of promotional tactics or incorporate all Code provisions and relevant WHA resolutions, allowing companies to manipulate the grey areas and gaps. For example, although the 2016 Guidance [80] clarifies GUMs as BMS and their promotion is prohibited, Dugro 3 was still promoted in the vlog and promotional ad banner in the parenting group in Myanmar. This is because the Myanmar law (adopted in 2014) covers up to 24 months of age [79], and Dugro 3 is unusually marketed as suitable for children from two to nine years to get around the law. In Pakistan, even though the law was amended in 2018, its scope for BMS products only covers up to 12 months of age [79], allowing companies like Nestlé to find a loophole to promote GUMs like Nactogrow. The Breast Milk Substitutes (Regulation and Control) Act of Kenya has provisions restricting incentives such as fellowships, study tours, and attendance at professional conferences, but there is no specific provision that restricts sponsorship of meetings of health professionals or scientific meetings [79]. Health claims are prohibited by resolution WHA 58.32 (2005) [28]. Vietnam’s Decree No. 100 bans advertising and other forms of promotion of BMS up to 24 months of age [79], the absence of specific health claims provisions was exploited by Colosbaby, as shown in the blatant claims on immunity. In order to advertise, Colosbaby also bypassed the law by falsely declaring their BMS products as ‘dietary supplements’ in the official product declaration process (D. Vu, personal communication, September 11, 2020). Article 6.8 of the Code places restrictions on donations of equipment, and according to the 2016 Guidance, donations of equipment to health facilities and giving gifts or incentives to health staff are prohibited to avoid conflicts of interest [28]. Burkina Faso adopted a national decree to restrict BMS promotion in 1993. It is moderately aligned with the Code, leaving gaps in areas such as donations of equipment to the health systems [79].

Many a time, even with laws that do give effect to the Code, the monitoring and enforcement are lagging. For example, the Milk Code in the Philippines prohibits donations of BMS products. However, enforcement is weak at local levels [79,88] and donations were rampant during the COVID-19 pandemic. The Milk Code also restricts inducement for health professionals that create conflicts of interest, such as funding for meetings, seminars, or conferences [79]. However, the lag in enforcement has allowed these activities to reach health professionals. Even though Indonesia has relatively strict provisions that are moderately aligned with the Code, there are gaps and overlapping areas among the laws, with little to no enforcement despite systematic and continuous violations [89]. During data collection of this study, it was also found that many countries do not have systematic national monitoring, and the monitoring findings sometimes do not reach relevant policymakers and enforcement agencies.

Logistical barriers such as ineffective communication and coordination among government agencies, lack of monitoring guidelines, absence of clearly specified sanctions and incoherent policies also hinder enforcement. For instance, in the Philippines, a House Bill was transmitted to the Senate during the COVID-19 pandemic that allows private sector donations of products and services (including BMS) for disaster relief [90], contradicting the aim and functions of the Milk Code. 

A huge threat to effective Code implementation is the conflating of company voluntary monitoring and self-assessed compliance with the necessity for legal measures and independent monitoring. A critical review of company internal compliance manuals (e.g. Nestlé and Danone) and other corporate governance, risks and compliance (GRC) statements (e.g. Mead Johnson) that claim to support Code compliance has shown that company interpretations of the Code are always set at a much lower benchmark compared to those by global health authorities such as WHO and UNICEF. These ‘subpar’ interpretations include: compliance with the Code is only necessary in developing countries, the scope only covers products up to six months of age, and adjusting level of Code adherence by dividing the world into high risk and low risk countries [30,31]. These initiatives help companies create good image, they can also be tools to help companies circumvent the Code and national laws. Initiatives such as the recent BMS Call to Action [91], one that mobilizes BMS companies to achieve Code compliance voluntarily by 2030, can be seen as a fresh approach to appeal to the goodwill of companies. However, the partnership with companies is a departure from the Code and the commercially-influenced monitoring is fraught with conflicts of interest [92]. In addressing this initiative, companies have demonstrated discrepant interpretation of the Code, such as willingness to only comply with the Code selectively; conflating Code compliance with interfering with women’s right to choose; assuming the role to provide education and other forms of support for health workers; and choosing to only abide by national laws when there are gaps between the Code and national laws [91]. Simply the countless violations documented through the years should make a compelling argument that legal measures and independent monitoring cannot be replaced by other voluntary measures and compliance claims.

## 6. Limitations

The data of the qualitative part of this study were collected through purposive and convenience sampling; therefore, the findings were not representative in covering an exhaustive range of marketing activities. Data collection took place during the COVID-19 pandemic, and many of the entries were collected from internet searches, which revealed little on what was happening in other sites such as shops and healthcare facilities. The analyses used cross-sectional data to obtain descriptive results, limiting the ability to make generalizations, predictions, and causal relations (e.g., this study is not able to provide insight on whether donations or social media promotion is more responsible for suboptimal breastfeeding). The quantitative data were only limited to the Philippines, and thus unable to provide a comprehensive picture of wider marketing trends. The crowd-based reporting was not exhaustive, and thus not a representative sample of inappropriate marketing practices in the country. Future research can explore the relationship between the prevalence of aggressive marketing and Code implementation status, the impact of public awareness of the Code on marketing practices, the impact of BMS marketing on consumer views and behaviors, and longitudinal studies on factors that influence overall Code implementation and company behavior over time.

## 7. Recommendations and Conclusions

### 7.1. Immediate Action for Governments and NGOs

Adopting laws and strengthening enforcement take time. There are interim and immediate actions that can lessen the harm caused by unregulated marketing practices:

#### 7.1.1. (i) Monitoring Findings to Be Used to Inform Actions at the WHA

Targeted monitoring should be mobilized immediately (see Section 7.2 for more on monitoring), and findings should be used to inform actions in the upcoming WHA meetings and agenda, particularly ways to protect breastfeeding in the context of COVID-19. Governments and NGOs should advocate for stronger and continued Code implementation and support at the national level at the meeting planned for the 40th anniversary of the adoption of the Code in the Seventy-fourth WHA in 2021. The results from monitoring the digital platforms should be used to inform the upcoming Report on the Scope and Impact of Digital Marketing Strategies for BMS Promotion in the Seventy-fifth WHA in 2022 [86]. The findings should also be used to urge the WHA to maintain ongoing reporting on Code issues during WHA meetings.

#### 7.1.2. (ii) Targeted Enforcement Can Be Conducted with Existing Inspections

Countries that have adopted laws based on the Code, but are lagging in enforcement need to start enforcing. Enforcement does not need to be large-scale operations that tackle all kinds of violations all at once. It can be targeted and integrated into regular and existing inspections, tailored according to capacity and existing marketing trends. Marketing practices are mostly centrally-planned and many practices are the same nationwide, duplication of efforts should be avoided. Labeling and promotion at retail outlets and on digital platforms are more visible targets with which to start. Taking concrete action on violations that are easier to detect will show companies that the government is serious about enforcement, setting precedents to deter others from violating the law. Enforcement agencies and officials can gain experience in more visible areas before proceeding to complicated areas that require more investigation, such as promotion in the health systems. In some cases, enforcement action can be as simple as issuing warning letters and providing a period for corrective action—while others require more serious penalties and sanctions. For countries with no laws, Code monitoring can be conducted in the same targeted approach. Find viable ways to publicize results and use them to advocate for the adoption of legal measures.

#### 7.1.3. (iii) Breastfeeding in the Context of COVID-19 Should Be Promoted

It is important to immediately address the misinformation about breastfeeding and COVID-19 to minimize opportunities companies exploit to incite unfounded fear around transmission and safety issues. Governments (including health systems) and NGOs need to scale up the effort on public awareness campaigns to reassure the public that breastfeeding with appropriate precautions is not only safe, but necessary for the benefits of health in this dire global crisis.

#### 7.1.4. (iv) Efforts Must Be Made to Prevent Spillover Effect of BMS Donations

COVID-19 has provided an entryway for rampant donations that can severely undermine breastfeeding practices. It is important to reach out to major emergency and community programs (e.g., food pantries and social services programs) to sensitize them to the harmful effects of donations and inappropriate distribution of BMS products on child health. Practices in line with the OG-IFE, such as allocating public funds to support proper procurement of BMS, and providing support for programs on needs assessment and appropriate distribution of BMS products, should be explored. There should be an increased effort on raising public awareness of the harmful effect of BMS donations.

#### 7.1.5. (v) Code-Compliant Best Practices in Health Systems and Programs Need to Be Established as Policy 

Companies often target health workers and health systems. Regardless of national legislation, healthcare facilities need to develop best practices policies that are free from conflicts of interest and are compliant with the Code to be followed by all workers. Basic Code knowledge and the importance of Code compliance should be included in health-related education curriculum and health workers’ training.

### 7.2. Long-Term Action on Code Implementation

#### 7.2.1. (i) Code Awareness to Be Raised Through Public Education and Social Messages

Though there has been much progress in popularizing the Code to the public in recent years, it is still not well-understood by the general public. This lack of understanding can easily benefit BMS companies—instead of being understood as recommendations to protect children from harmful marketing practices, the Code was misconstrued in the media as contributing to child hunger by restricting donations during the pandemic [93]. It is critical to promote the social message that the Code concerns everyone, apart from protecting babies, it also protects the public’s right to accurate and adequate health information, and that it is the governments’ duty to protect the avenues of disseminating such information. Public awareness initiatives that connect the dots between breastfeeding, inappropriate marketing, and the Code are needed, especially in the context of COVID-19. These initiatives should target multiple sectors of the public using popular media platforms, aiming to reinstate breastfeeding as the cultural norm, de-normalize all BMS promotion, and denounce companies and media platforms when they violate the Code.

#### 7.2.2. (ii) Monitoring Must Be Systematic, Sustainable, and Independent

Riding on the momentum of public education, sustainable and periodic community monitoring should be mobilized. For monitoring to reveal an accurate picture of marketing activities, it needs to be independent and free from industry influence. At the global level, IBFAN has demonstrated a community-based voluntary Code monitoring model in the last few decades. Each country is unique and thus needs to assess what is manageable and put that into action, no matter how small the scale. The Philippines, with its affordable web-based and crowd-sourced methods, sets an example for sustainable national monitoring. Government should engage with NGOs in related fields (e.g., public health, child nutrition, women and children welfare, and consumer rights) for more strategic and accessible monitoring. As marketing tactics are always ahead of national laws and regulations, monitoring results should be communicated to policymakers, and if necessary, be used to advocate for stronger Code implementation (e.g., reviewing legal measures that are not up to date to capture new marketing trends) and enforcement.

#### 7.2.3. (iii) Social Media Platforms Need to Be Held to Account for Violating the Code

Although promotion is rife on the digital platform, especially social media, it has also made monitoring in some ways more convenient. Monitors may organize themselves to present violations collected from social media and demand these platforms to act in compliance with the Code. Options include requesting social media outlets to incorporate the Code into their internal vetting policy, provide user features for reporting Code violations, and take down posts that violate the Code.

#### 7.2.4. (iv) Laws Need to Be Reviewed (or Adopted) to Keep Up with Marketing Tactics

The findings in this study showed how companies take advantage of gaps between what national regulations restrict and what companies can do given their reach, power, and blatant disregard of the Code. Monitoring findings, together with relevant WHA resolutions, can be used as guidance for review of existing laws or adopting new laws. Though governments are obligated to adopt the Code into legal legislation as minimum requirement, they are encouraged to go further with more stringent measures to cover a wider range of marketing tactics. 

#### 7.2.5. (v) Conflicts of Interest Hamper Political Will, and Thus Need to Be Tackled

Conflicts of interest within the government and health systems, such as sponsorship of public health programs, industry’s participation in policymaking and monitoring, and powerful industry lobbying can contribute to a lack of political will that impedes the adoption of strong laws and proper enforcement. Realistic compliance and management framework and policy must be developed and implemented to avoid conflicts of interest at the institutional level, as well as to guide officials to take personal responsibility to identify and resolve problematic situations. Training and capacity building should be provided to sensitize officials to the risks of undue influence have on their duty to protect health.

#### 7.2.6. (vi) Alternative Funding Options in Health Systems Should Be Explored

Health workers, including managers and administrators, should not accept resources and funding that are linked to BMS companies. Governments and policymakers can explore direct or indirect systematic taxation to support health workers’ training, research activities, and public health programs (e.g. imposing tax on profits from BMS sales and using that to support scientific meetings, seminars, and research).

#### 7.2.7. (vii) High-level Political Support Must Be Cultivated

While it is feasible for monitoring to take place at a community level, effective Code implementation requires drafting and adopting strong legislation, and enforcing it. For that, high-level political support at a national level is critical. Mobilizing support from NGOs, health professionals, government officials from relevant departments, policymakers, and intergovernmental agencies is essential for a concerted and multi-sector approach. It is vital to facilitate effective communications, collaboration, and coordination among government departments and other responsible parties.

### 7.3. Conclusions

The overarching concept of consumer vulnerability is encompassed by the themes emerged from the findings. The devastating effects of COVID-19 on the healthcare, economic, food, education, and social support systems have put many people in vulnerable positions in multifarious ways. Tactics including unfounded health claims and misguided information on breastfeeding are designed to cultivate parents’ fear and uncertainty. This makes them susceptible to not just the BMS products, but also the inherent marketing messages: the sense of reassurance found in the idea of immunoprotection. Paradoxically, companies also capitalize on people’s sense of hope and solidarity that is born out of the pandemic. The donations campaigns and offering of support and services have a solidarity effect that allows companies to appear as supporters or even comrades in the fight against COVID-19. This helps companies gain goodwill, a valuable marketing asset. Companies take advantage of the vulnerability inherent in these sentiments through emotional appeals. As people spend more time on digital platforms, their personal data become more accessible to advertisers for marketing purposes. Economic downturns have caused financial hardships to many families. Companies target vulnerable populations, including low-income families, with free samples and sales discounts linked to COVID-19. The lack of public awareness about how BMS donations harm public health has ‘empowered’ companies to continue to reinvent this old trick—packaging them into charitable initiatives through partnerships with foundations and NGOs.

Beyond the pandemic, decades of aggressive marketing, inadequate maternity protection, and scarce breastfeeding support have enabled formula and bottle-feeding to become a widely-accepted social norm, with which breastfeeding has to compete. The stark difference in resources between BMS companies and those available for protecting and supporting breastfeeding places women and children in a kind of structural vulnerability —one that undermines women’s confidence in their ability to breastfeed, the public’s access to accurate health information, and children’s right to optimal health.

Breastfeeding is a core part of optimal nutrition in the first 1000 days, it is also a natural health equalizer that gives a child who was born into poverty a fairer start. The health benefits persist until later in life into adulthood. Improvement in breastfeeding practices needs multi-level interventions and Code implementation alone is not enough. However, it is a critical public health instrument to help build an enabling environment where breastfeeding can even stand a chance. Although many of the tactics in our findings are not new, this study has provided evidence that companies are exploiting a global pandemic as a new marketing entry-point. The fact that inappropriate marketing can even thrive in global emergencies, indicates that companies are nefariously taking advantage of the lagging Code implementation and enforcement. The imminent risks of increased child mortality, morbidity, and malnutrition during the COVID-19 pandemic should convey to governments the urgency to drastically scale-up efforts to restrict harmful marketing practices of BMS companies to protect breastfeeding.

## Figures and Tables

**Figure 1 ijerph-18-02381-f001:**
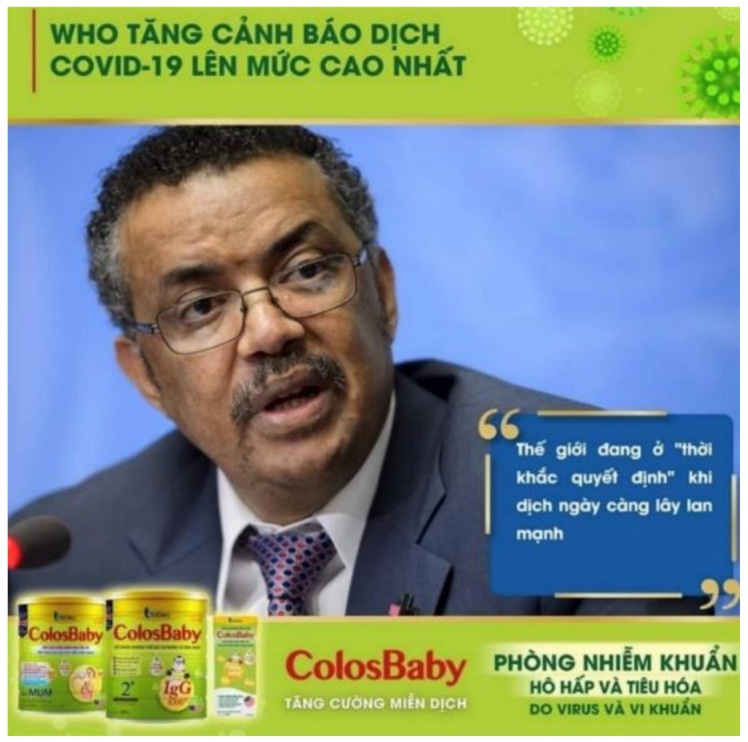
A screenshot from Facebook of Colosbaby advertisement in Vietnam that featured the World Health Organization’s (WHO) Director-General headshot with a caption at the top, “WHO raises COVID-19 threat warning to its highest level”. “The world is entering a decisive moment when coronavirus disease is spreading more rapidly” appeared like a quote from him. Next to the products were captions “ColosBaby Boost immune system” and “Prevent respiratory and digestive infections caused by viruses and bacteria” (translated from Vietnamese).

**Figure 2 ijerph-18-02381-f002:**
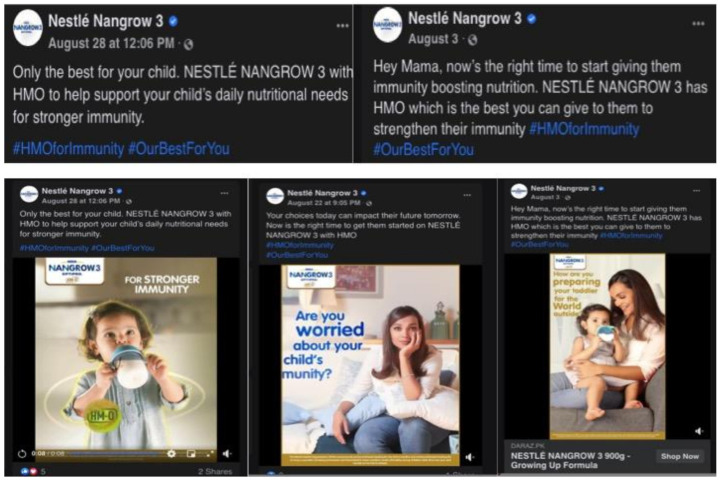
Screenshots of advertisements from Nestlé’s Nangrow 3 Facebook page in Pakistan, featuring immunity claims and fear-provoking questions.

**Figure 3 ijerph-18-02381-f003:**
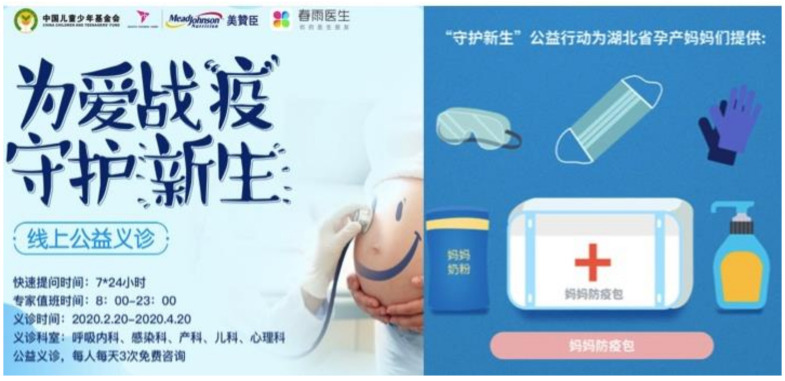
Screenshot of Mead Johnson’s campaign from the China Children and Teenager Fund’s website, with the caption “Conquer virus for love to protect new lives” (translated from Chinese). Image on the right shows items in donation packets, including milk for mothers.

**Figure 4 ijerph-18-02381-f004:**
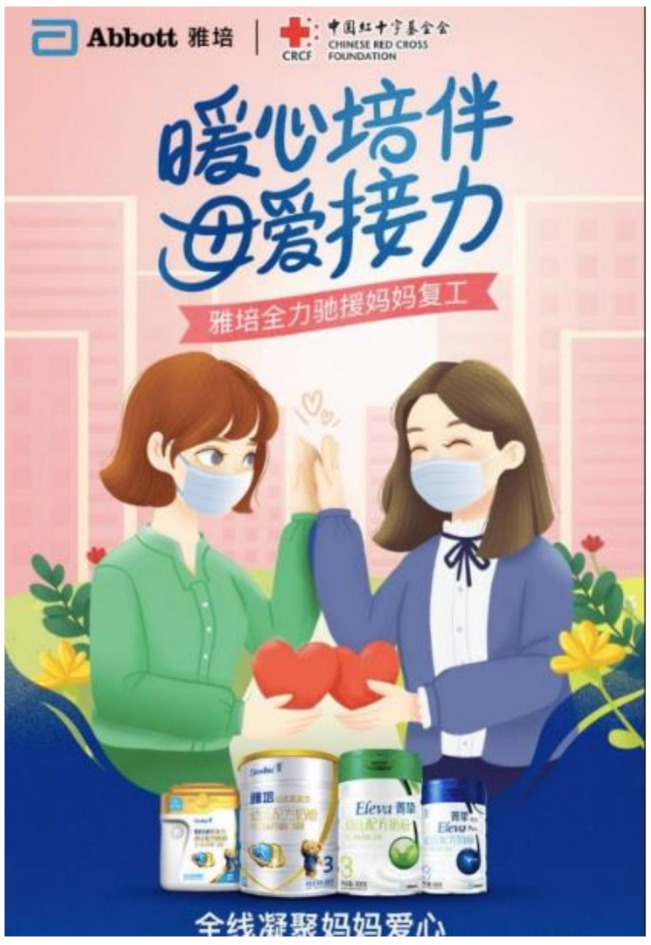
A screenshot from an online news portal in China of Abbott’s campaign with the Chinese Red Cross Foundation. The e-poster featured the slogan “A warm heart to keep you company and to carry on with the love of a mother: Abbott fully supports mothers returning to work” (translated from Chinese) and BMS products.

**Figure 5 ijerph-18-02381-f005:**
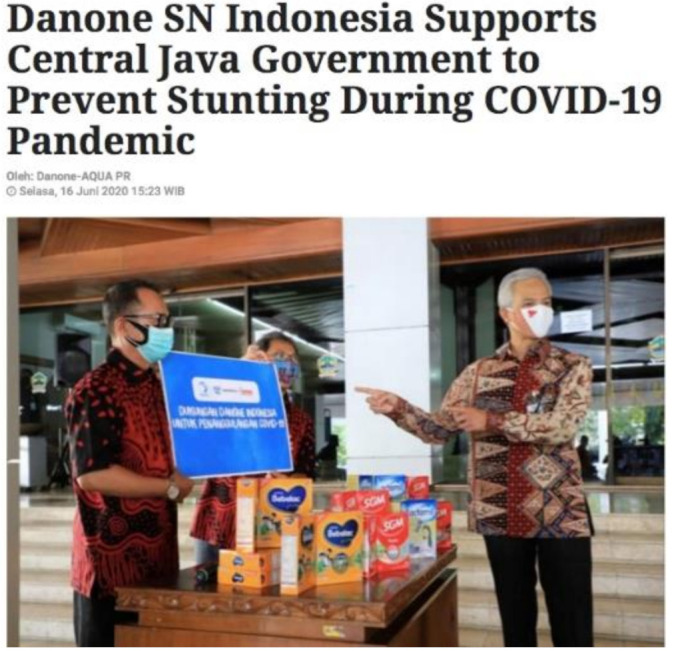
A screenshot of a news article about Danone’s donation of formula products (including Bebelac and SGM) to Central Java government in Indonesia.

**Figure 6 ijerph-18-02381-f006:**
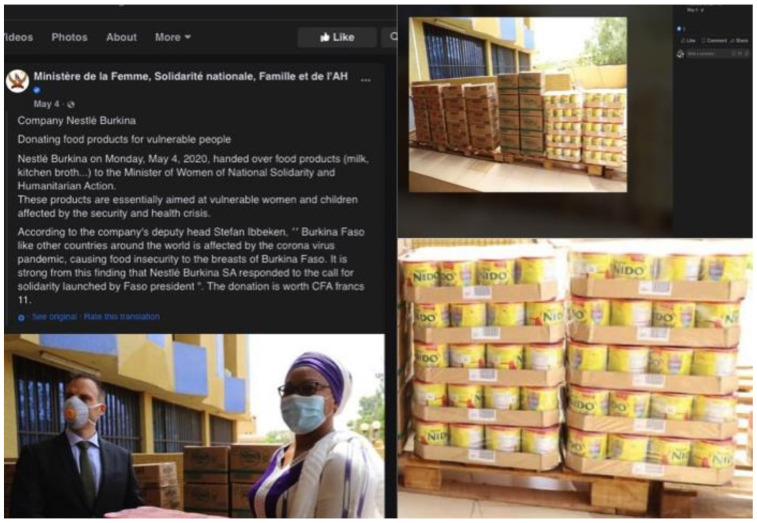
Screenshots of Nestlé’s donations to the Ministry of Women, National Solidarity and Family, and Ministry of Humanitarian Action in Burkina Faso taken from Facebook.

**Figure 7 ijerph-18-02381-f007:**
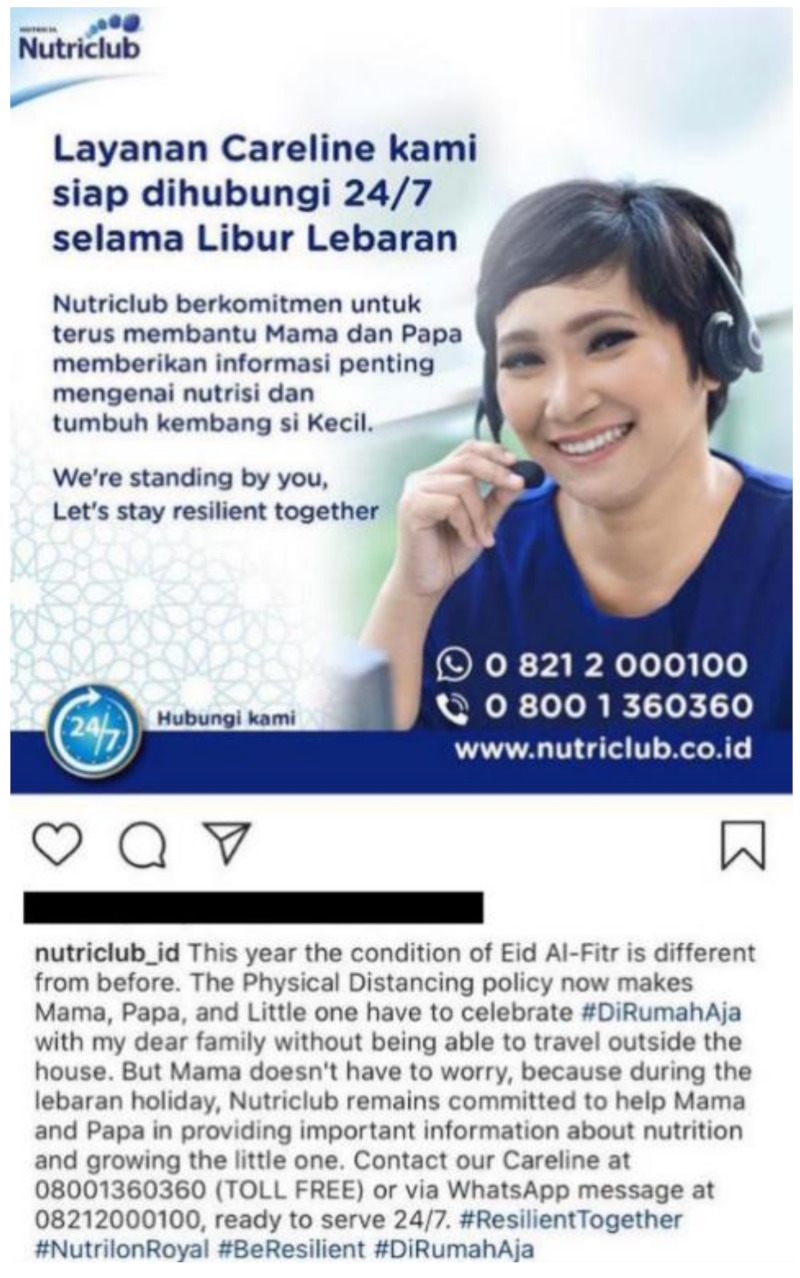
Screenshot of Danone’s Nutriclub Indonesia taken from Instagram, with the headline “Our careline services are available 24/7 during the Eid holidays” (translated from Indonesian).

**Figure 8 ijerph-18-02381-f008:**
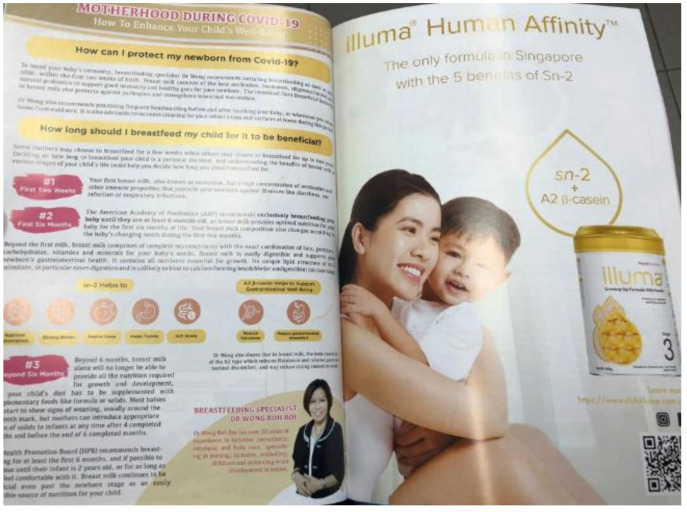
Image of Nestlé’s advertisement of Illuma 3 from Mummys Market Pregnancy and Baby Guide in Singapore. Next to the advertisement was ‘information’ that highlighted “motherhood during COVID-19” and breastfeeding.

**Figure 9 ijerph-18-02381-f009:**
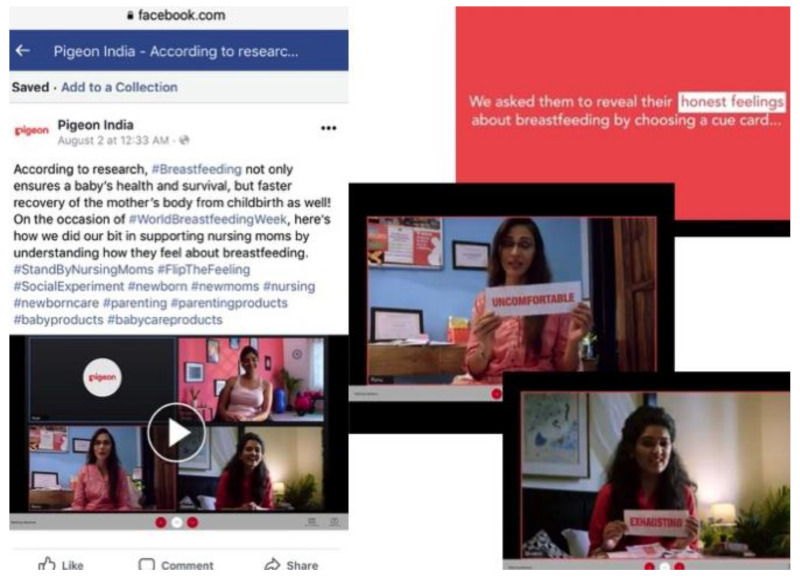
Screenshots of Pigeon India’s #standbynursingmoms campaign video from Facebook, with women revealing their “honest feelings” about breastfeeding as “uncomfortable” and “exhausting”.

**Figure 10 ijerph-18-02381-f010:**
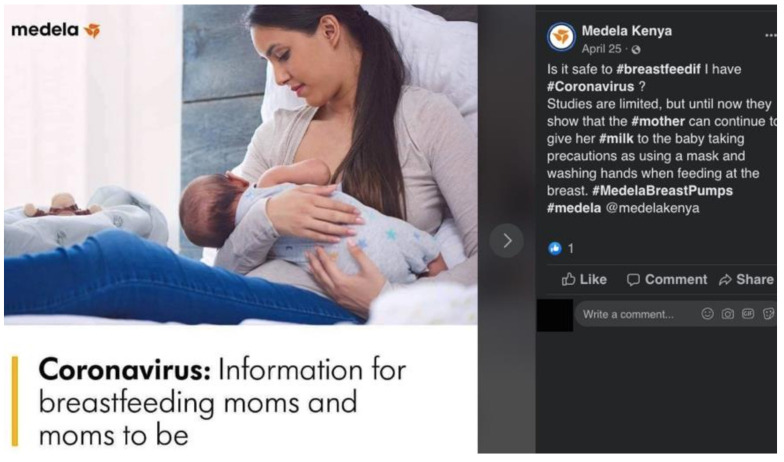
Screenshot of Medela Kenya’s Facebook post on “Coronavirus: Information for breastfeeding moms and moms to be”. The “information” emphasized “giving milk” without directly mentioning “breastfeeding”. Hashtags included “#MedelaBreastPumps”.

**Figure 11 ijerph-18-02381-f011:**
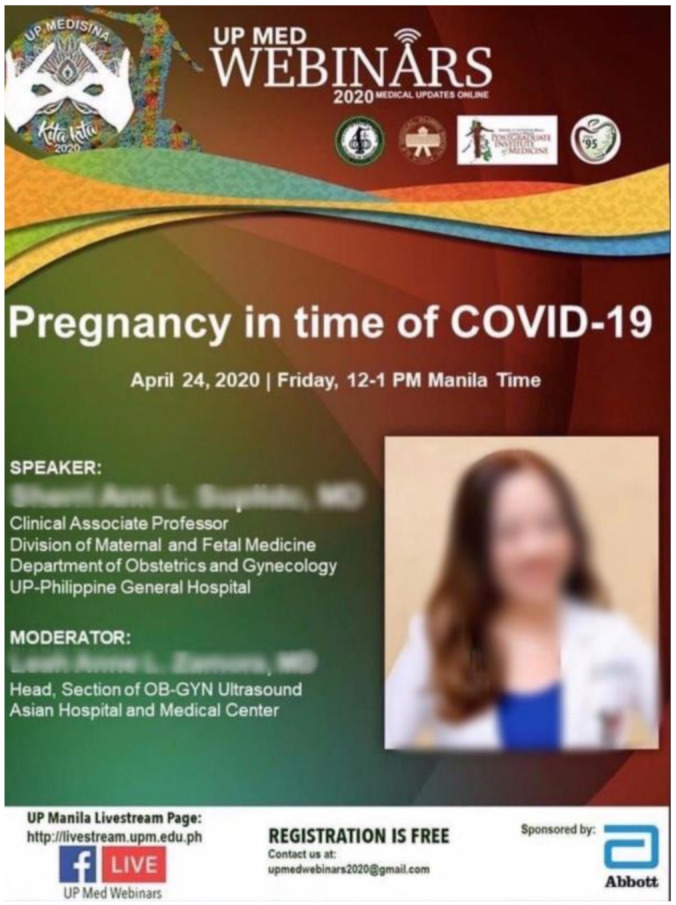
Screenshot of a webinar on pregnancy and COVID-19 organized by Abbott and the Medical College of the University of the Philippines from Facebook.

**Figure 12 ijerph-18-02381-f012:**
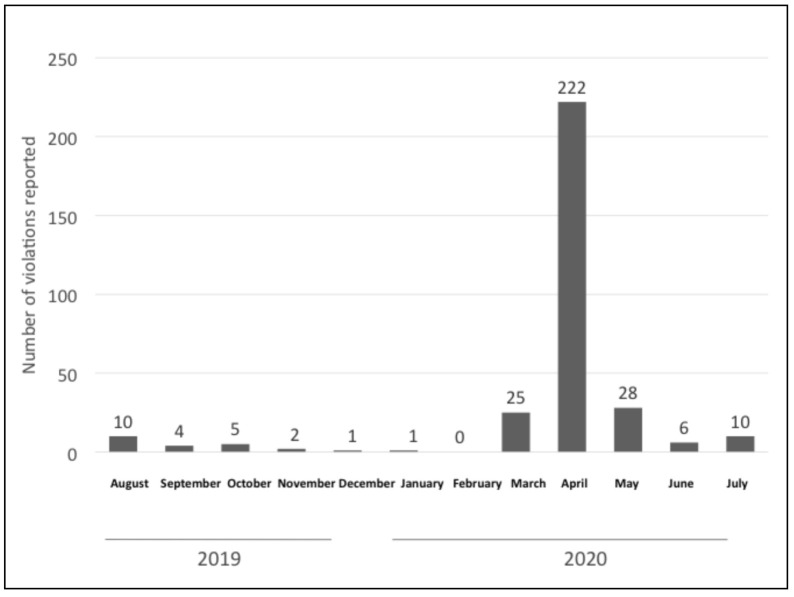
Number of reported violations in the Philippines (January 2019 to July 2020).

**Figure 13 ijerph-18-02381-f013:**
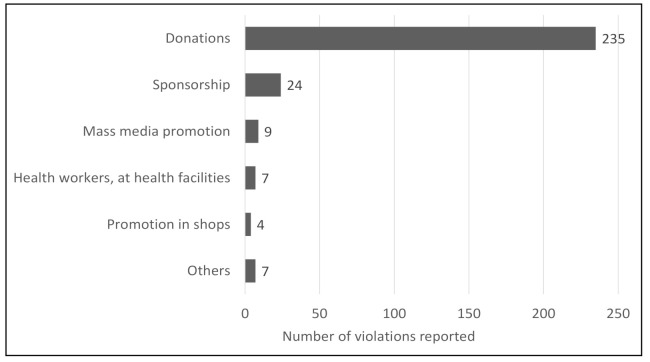
Percentage of reported violations in the Philippines since the pandemic started.

**Table 1 ijerph-18-02381-t001:** Summary of marketing themes, companies, countries, and where promotion was found.

Theme	Company	Country	Where Activity/Promotion Was Found
i. Unfounded health claims on immunity that prompt fear	Vitadairy	Vietnam	Online shopping portal
	Nestlé	Pakistan	Social media (Facebook)
ii. Association with public health authorities to gain legitimacy	Vitadairy	Vietnam	Social media (Facebook)
iii. Appeal to public sentiment on solidarity and hope	Reckitt Benckiser	China	Partner-NGO website
	Abbott	China	Online news portal
	FrieslandCampina	China	Online news portal
iv. Influx of donations of BMS products and supplies related to COVID-19	Danone	Indonesia	Online news portal
	Nestlé	Canada	Company website
	Feihe	China	Online news portal
	Danone	Malaysia	Company website
	Nestlé	Burkina Faso	Social media (Facebook)
	Nestlé	Burkina Faso	Online news portal
v. Prominent use of digital platforms to reach out to parents	Reckitt Benckiser	China	Partner-NGO website
	Feihe	China	Online news portal
	Danone	Indonesia	Social media (Instagram)
	Danone	Myanmar	Social media/Vlog (Facebook)
	Danone	Myanmar	Social media/Parenting group (Facebook)
vi. Promoting uncertainty through endorsing breastfeeding	Nestlé	Singapore	Print magazine
	Pigeon	India	Social media (Facebook)
	Medela	U.S.	Company website
	Medela	Kenya	Social media (Facebook)
	Danone	Laos	Social media (Facebook)
vii. Discounts on BMS products that are linked to COVID-19	Nestlé	U.S.	Company website
	Abbott	U.S.	Company website
viii. Reaching out to health professionals through sponsoring educational events on topics relating to COVID-19 and infant and young child feeding	Nestlé	Kenya	Public health website (blog)
	Abbott	The Philippines	Social media (Facebook)

## Data Availability

Data in the qualitative section (images of online content) are embedded in the paper and available as Appendix A. Hyperlinks to the original internet source where images were obtained are included in the References section. Web archive links are available from the corresponding author upon reasonable request. The quantitative dataset is not readily accessible for the public but can be obtained after an official request is approved by the Department of Health, Philippines.

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
