# Peer review of "Old Tricks, New Opportunities: How Companies Violate the International Code of Marketing of Breast-Milk Substitutes and Undermine Maternal and Child Health during the COVID-19 Pandemic"

_ijerph, 2021, doi:10.3390/ijerph18052381_

Round 1

Reviewer 1 Report

Summary and general comments

This paper presents the results of a study aimed to identify marketing tactics of BMS companies during the COVID-19 pandemic. The study was conducted in 14 countries from August to October 2020, throuhg internet, print magazines, networks of health officials and professionals, health facilities, and shops.   The authors analysed marketing themes and trends on marketing. The topic of the study concern an important subject for breastfeeding promotion, protection and support, specifically to the compliance of the International Code of Marketing of Breastmilk Substitutes, addressing the importance to regulate breastmilk substitutes marketing in general, but particularlly, in an emergy context such as the current COVID-19 pandemic. This is a well written paper, and it is of paramount importance to document breast-milk subtistutes manufactures´ unethical marketing strategies.  The main minor comment to the authors is that discussion section, even thouhg has very well document arguments, is sometimes repetitive with the result section. Minor comments by section are described below:

Minor comments by sections

Aim, Material and Methods section:

Page 3, lines 126-136: The objective of the paper and research questions do not correspond. The objective does not reflect all the thorough approach that the authors had. It is suggested that authors include only either the objective (including the research questions as part of the objectives) or the research questions.

Page 4 Line 158-177: Somewhere within these lines, specify how many researchers/research assistants/fieldworkers did the coding and indicate the validity and inter-rate reliability.

Discussion section:

Even thouhg has very well document arguments, is sometimes repetitive with the result section.

Recommendations and conclusions section:

Agree with all author´s recommendation. However, all of them are more  long term recommendations. Is there any thing that can be recommended to stop sooner the unethical marketing? Particularlly in an emergency context, such as the CODIV-19.

Figures and tables:

All figures (including  supplementary material) are shocking. It would add more, if authors include the English translation (when needed) not only as body text (as it is now), but as figure note.

Author Response

Dear Reviewer,

Thank you very much for the helpful comments and suggestions for our revisions. Also thank you for the encouraging and reassuring words. 

Overall, following your suggestions, we have re-organized and shortened certain parts to improve on the flow, especially in the 'Discussion' and 'Recommendations & Conclusions' sections. Below is our point-by-point response:

Minor comments by sections

Aim, Material and Methods section:

Page 3, lines 126-136: The objective of the paper and research questions do not correspond. The objective does not reflect all the thorough approach that the authors had. It is suggested that authors include only either the objective (including the research questions as part of the objectives) or the research questions.

Response:

We revised as suggested to make the aim and research questions consistent.

Page 4 Line 158-177: Somewhere within these lines, specify how many researchers/research assistants/fieldworkers did the coding and indicate the validity and inter-rate reliability.

Response:

We added the information as suggested.

Discussion section:

Even though has very well document arguments, is sometimes repetitive with the result section.

Response:

We revised to shorten the text and minimized the repetition.

Recommendations and conclusions section:

Agree with all author´s recommendation. However, all of them are more long term recommendations. Is there anything that can be recommended to stop sooner the unethical marketing? Particularly in an emergency context, such as the CODIV-19.

Response:

We revised the recommendations and conclusions. Recommendations are now divided into 'Immediate' and 'Longer-term' actions. Immediate actions include using monitoring findings to inform upcoming WHA actions (2021 and 2022), for countries that have laws to start 'targeted' enforcing, and addressing misinformation on breastfeeding in the context of COVID-19.

Figures and tables:

All figures (including supplementary material) are shocking. It would add more, if authors include the English translation (when needed) not only as body text (as it is now), but as figure note.

Response:

We added the suggested translated quotes in English. 

Reviewer 2 Report

The authors effectively describe their findings on the violations of the International Code which have been seen during the period of the COVID 19 pandemic on the part of manufacturers of infant formula (Breast Milk Substitutes). The source of data is the internet and social media posting and no direct research has been conducted with consumers which could be seen as a limitation. Violations of the code in the Philippines have been reported though it it interesting that most of these occurred at the beginning of the pandemic, and fewer during the later periods.

The paper is well written though lengthy and provides new information on the way that formula companies have taken advantage of a serious health crisis to promote their products and in so doing, adversely influence the health of infants and young children.

Author Response

Dear Reviewer,

Thank you very much for the helpful comments and suggestions for our revisions. Also thank you for the encouraging words. Overall, following your suggestions, we have re-organized and shortened certain parts to improve on the flow, especially in the 'Discussion' and 'Recommendations & Conclusions' sections. Below is our point-by-point response:

The authors effectively describe their findings on the violations of the International Code which have been seen during the period of the COVID 19 pandemic on the part of manufacturers of infant formula (Breast Milk Substitutes). The source of data is the internet and social media posting and no direct research has been conducted with consumers which could be seen as a limitation. Violations of the code in the Philippines have been reported though it it interesting that most of these occurred at the beginning of the pandemic, and fewer during the later periods.

Response:  Thank you very much. Yes, we agree data from interviews with consumers regarding marketing tactics and how the tactics affect their views and decisions would enrich the paper in a substantial way. We had a short time line to work with - data collection took a huge chunk, and did not have enough time to go through the ethical review process. 

The paper is well written though lengthy and provides new information on the way that formula companies have taken advantage of a serious health crisis to promote their products and in so doing, adversely influence the health of infants and young children.

Response:

We shortened the text in the manuscript and re-arranged certain parts to improve on the flow in the Discussion and Recommendations sections. 

Reviewer 3 Report

General: This is a very timely and considered piece of research. There is a little repetition between results and discussion as all the themes needed individual attention making it quite long. It is a shame the Code violations research was limited to one country as it hampers our understanding in a wider context but this is mentioned in the limitations. Overall a very interesting and detailed piece of work and will likely be useful for policy makers.

Minor comments: Line 39 I would suggest removing intelligence as a factor as not a direct health outcome and still hard to reliably cite without more detail.

Line 87 - The Code has human rights implications...perhaps elaborate on this sentence to explain how...

Line 142-145 - perhaps need to explain why these countries included and not others?

Table 1 - I would relabel the themes in the table to Roman numerals like in the text or change it the other way around to standardise the use of the 1-8.

Line 496-7 - re-write sentence as is clumsy to read

Author Response

Dear Reviewer,

Thank you very much for the helpful comments and suggestions for our revisions. Also thank you for the encouraging and reassuring words. 

Overall, following your suggestions, we have re-organized and shortened certain parts of the text to improve on the flow, especially in the 'Discussion' and 'Recommendations & Conclusions' sections. Below is our point-by-point response:

General: This is a very timely and considered piece of research. There is a little repetition between results and discussion as all the themes needed individual attention making it quite long. It is a shame the Code violations research was limited to one country as it hampers our understanding in a wider context but this is mentioned in the limitations. Overall a very interesting and detailed piece of work and will likely be useful for policy makers.

Response:

Re: the quantitative data - Yes it is a shame that we only had one country's quantitative data, which limits the study. We tried to collect quantitative data from other countries but none except for Philippines has systematic monitoring that provides 'workable' numbers that can be used in a research. 

Minor comments: Line 39 I would suggest removing intelligence as a factor as not a direct health outcome and still hard to reliably cite without more detail.

Response:

We revised as suggested.

Line 87 - The Code has human rights implications...perhaps elaborate on this sentence to explain how...

Response:

We revised as suggested.

Line 142-145 - perhaps need to explain why these countries included and not others?

Response:

Revision was made as suggested. The geographical 'scope' of our data collection was mostly based on our professional network, and we only selected examples of violations that made reference to or coincided with COVID-19 for further analysis. 

Table 1 - I would relabel the themes in the table to Roman numerals like in the text or change it the other way around to standardise the use of the 1-8.

Response:

We revised as suggested.

Line 496-7 - re-write sentence as is clumsy to read

Response:

We revised as suggested.

This manuscript is a resubmission of an earlier submission. The following is a list of the peer review reports and author responses from that submission.